# Alteration of the Neuromuscular Junction and Modifications of Muscle Metabolism in Response to Neuron-Restricted Expression of the CHMP2B^intron5^ Mutant in a Mouse Model of ALS-FTD Syndrome

**DOI:** 10.3390/biom12040497

**Published:** 2022-03-24

**Authors:** Robin Waegaert, Sylvie Dirrig-Grosch, Haoyi Liu, Marion Boutry, Ping Luan, Jean-Philippe Loeffler, Frédérique René

**Affiliations:** 1INSERM U1118 Mécanismes Centraux et Périphériques de la Neurodégénérescence, Centre de Recherche en Biomédecine de Strasbourg, Université de Strasbourg, 67000 Strasbourg, France; robinwaegaert@gmail.com (R.W.); grosch@unistra.fr (S.D.-G.); hy.liu0731@gmail.com (H.L.); marionboutry@orange.fr (M.B.); loeffler@unistra.fr (J.-P.L.); 2Medical Center, Shenzhen University Health Science Center, Shenzhen 518060, China; luanping@szu.edu.cn

**Keywords:** CHMP2B^intron5^, motor neuron disease, neuromuscular junction, neurophysiological evaluation, motor phenotype

## Abstract

CHMP2B is a protein that coordinates membrane scission events as a core component of the ESCRT machinery. Mutations in CHMP2B are an uncommon cause of amyotrophic lateral sclerosis (ALS) and frontotemporal dementia (FTD), two neurodegenerative diseases with clinical, genetic, and pathological overlap. Different mutations have now been identified across the ALS-FTD spectrum. Disruption of the neuromuscular junction is an early pathogenic event in ALS. Currently, the links between neuromuscular junction functionality and ALS-associated genes, such as CHMP2B, remain poorly understood. We have previously shown that CHMP2B transgenic mice expressing the CHMP2B^intron5^ mutant specifically in neurons develop a progressive motor phenotype reminiscent of ALS. In this study, we used complementary approaches (behavior, histology, electroneuromyography, and biochemistry) to determine the extent to which neuron-specific expression of CHMP2B^intron5^ could impact the skeletal muscle characteristics. We show that neuronal expression of the CHMP2B^intron5^ mutant is sufficient to trigger progressive gait impairment associated with structural and functional changes in the neuromuscular junction. Indeed, CHMP2B^intron5^ alters the pre-synaptic terminal organization and the synaptic transmission that ultimately lead to a switch of fast-twitch glycolytic muscle fibers to more oxidative slow-twitch muscle fibers. Taken together these data indicate that neuronal expression of CHMP2B^intron5^ is sufficient to induce a synaptopathy with molecular and functional changes in the motor unit reminiscent of those found in ALS patients.

## 1. Introduction

The neuromuscular junction (NMJ) is a specialized synapse where the nerve ending of a motor neuron (MN) establishes a chemical contact with a muscle fiber to control muscle contraction and govern vital processes such as breathing or voluntary movement. Reciprocal interactions between these two partners are required to maintain effective synaptic transmission and ensure the maintenance of a functional NMJ. Indeed, the pattern of MN firing will determine the contractile and metabolic characteristics of muscle fibers [1]. Disturbances in the communication within the motor unit lead to pathological situations as seen in motor neuron diseases, such as amyotrophic lateral sclerosis (ALS).

ALS is the most common motor neuron disease in adults. This neurodegenerative disease is characterized by a progressive degeneration of motor neurons, and severe skeletal muscle atrophy leading to paralysis and death by respiratory failure 2 to 5 years following the onset of symptoms [2]. To date, the pathogenesis of ALS remains poorly understood, however, it now clearly appears that disruption of the NMJ is an early event in ALS pathology. Indeed, studies in ALS patients showed that morphological changes at the neuromuscular junction in muscle biopsies and alterations in electrophysiological recordings of neuromuscular transmission are some of the earliest signs of disease [3,4,5,6]. Similar observations were performed in the mouse models of ALS, in which changes at the NMJ are visible before the appearance of clinical symptoms and precede the loss of MN cell body [5,7,8]. These data suggest that an impairment of neural drive, neuromuscular junction transmission, or excitation–contraction coupling, may participate to the initial steps of the disease. These alterations of the NMJs are associated with changes in muscle properties in patients and in mouse models of ALS [9,10,11,12,13,14].

Whether the early changes seen at the level of the NMJ/motor unit are solely due to altered MN physiology or also involve their target cells namely the skeletal muscle fibers remains an open question. Indeed, results obtained in mouse models are contradictory. It has been shown that expression of the SOD1 mutation limited to neurons induces an ALS-like disease with late onset characterized by motor neurons loss and muscle denervation [15,16]. In the converse situation where SOD1 mutation was expressed in a muscle-restricted manner, a destabilization of the NMJ has also been reported [17,18,19], suggesting that each partner could independently lead to NMJ destabilization.

ALS can be of sporadic or familial origin. Familial ALS (FALS) accounts for approximately 10% of all cases. Currently, mutations in more than 25 genes have been associated with ALS, with the *C9orf72* and *SOD1* mutations as the most common genetic causes [20]. Amongst these other genes, CHMP2B was the first gene associated with both ALS and frontotemporal dementia (FTD) [21,22,23,24,25,26,27] showing for the first time a genetic link between these two neurodegenerative diseases. FTD, the second most common form of pre-senile dementia before the age of 65, is characterized by an atrophy of several brain structures, the frontal and temporal lobes of the cortex and hippocampus being the most altered [28]. 

CHMP2B, a member of the chromatin-modifying protein/charged multivesicular body protein family strongly expressed in the central nervous system [21], is part of ESCRT-III (endosomal sorting complex required for transport III), a complex involved in numerous biological events, some of which being particularly crucial for the development and functioning of the nervous system [29]. Indeed, this complex is involved in vesicles biogenesis by scission of preexisting membrane, and in neurons, ESCRT-III participates in vesicles trafficking, dendritic spines formation, and autophagy [30]. In 2005, the CHMP2B^intron5^ mutation was discovered in a large Danish family of FTD. This mutation is a missense mutation in the splice acceptor site of exon 6 generating two distinct aberrant transcripts, CHMP2B^intron5^ and CHMP2B^∆10^, both part of which encode proteins with a defective C terminus [21]. Since then, 11 new mutations in this gene have been found in ALS [22,24,26], FTD [21,23,25], and ALS–FTD [22] patients, supporting the idea of a pathological continuum. These mutations are disseminated throughout the sequence, irrespective of the considered pathology. However, the C-terminal region lacking in CHMP2B^intron5^ also contained point mutation sites found in ALS [22,24] and FTD [21,25] patients, suggesting a common contribution of this region to both pathologies. To date, two transgenic mouse lines expressing the CHMP2B^intron5^ mutant have been generated. Both develop a phenotype reminiscent of ALS and FTD with a progressive development of muscle weakness, alteration of motor functions, partial denervation of skeletal muscle and axonopathy in addition to behavioral changes [31,32,33]. Currently, it is not known whether CHMP2B^intron5^ mutation alters the functionality of the NMJ, and thereby induces changes in muscle properties as has been observed with mutants of other ALS-related genes. To address this question, we examined neuromuscular pathology in the transgenic line with neuron-restricted CHMP2B^intron5^ expression [32]. We used complementary approaches (behavior, histology, electroneuromyography, and biochemistry) to determine the extent to which neuron-specific expression of CHMP2B^intron5^ could impact the nerve-ending functionality and the skeletal muscle characteristics. We show that neuronal expression of the CHMP2B^intron5^ mutant is sufficient to trigger progressive gait impairment associated with structural and functional changes in the neuromuscular junction. Indeed, CHMP2B^intron5^ alters the pre-synaptic terminal organization and the synaptic transmission that ultimately lead to a switch of fast-twitch glycolytic muscles fibers to more oxidative slow-twitch muscle fibers.

Taken together these data indicate that neuronal expression of CHMP2B^intron5^ is sufficient to induce a synaptopathy with molecular and functional changes in the motor unit reminiscent of those found in ALS patients. 

## 2. Materials and Methods

### 2.1. Ethics Statement

All animal experimentations were performed in accordance with European regulations (Directive 2010/63/EU), institutional and national guidelines, and approved by the local ethical committee from Strasbourg University (CREMEAS) and the ministry of higher education and research, under numbers APAFIS#2255 and APAFIS#9494.

### 2.2. Animals

CHMP2B^intron5^ transgenic mice overexpressing the human CHMP2B^intron5^ mutant under the Thy1.2 promoter were obtained by breeding hemizygous (HE) mice carrying 6 copies of the transgene (mixed FVB/N-DBA/2-C57BL/6 background) and were genotyped as previously described [32]. HE mice were used in this study and compared with control non-transgenic (nTg) littermates. To allow direct visualization of motor neurons, HE-YFP mice were obtained by breeding HE CHMP2B^intron5^ mice with Thy1-YFP mice [34]. Mice were group-housed (2–5 per cage) in the animal facility of the medicine faculty of Strasbourg University, in a temperature- and humidity-controlled environment at 22 ± 1 °C under a 12-h light/dark cycle with unrestricted access to regular A04 rodent chow and water. 

### 2.3. Gait and Muscle Strength Analysis

Gait parameters of freely moving mice were measured using the CatWalk gait analysis system (Noldus Information Technology, Wageningen, The Netherlands) at 6, 12, and 18–24 months. This system provides accurate and repeatable measurements of gait function and spatial and temporal aspects of interlimb coordination. CatWalk instrument consists of a hardware system of a long, enclosed glass walkway plate, illuminated with green light, a high-speed video camera, and a software package for quantitative assessment of animal footprints. Prior to the experiment, mice were acclimatized and trained to voluntarily walk across the illuminated walkway in a dark and quiet room dedicated for behavioral experimentation. The recordings were carried out when the room was completely dark, except for computer screen. Each mouse was placed individually in the CatWalk walkway and allowed to walk freely, in an unforced manner and traverse from side to side the walkway glass plate. Mouse tracks that were straight without any interruption or hesitation were treated as successful runs. Runs with any wall climbing, grooming, and staying on the walkway were not analyzed. Three replicate crossings made by each mouse with at least five cycles of complete steps were analyzed with the CatWalk software. The base of support, the regularity index, and the duty cycle were analyzed. Base of Support (cm) is a static parameter corresponding to the mean distance between either front paws or hind paws. This parameter was shown to be altered after trauma of the spinal cord [35]. Regularity index (%) is a fractional measure of inter-paw coordination used as a measure of the degree of interlimb coordination during gait [36]. It expresses the number of normal step sequence patterns (NSSP) relative to the total number of paw placements (PP). In healthy, fully coordinated animals, its value is 100%. The formula to calculate the regularity index is: Regularity Index = (NSSP × 4)/PP × 100. Duty Cycle expresses stand (time of contact of a paw in seconds during a step cycle) as a percentage of step cycle duration where the step cycle is the time in seconds between two consecutive initial contacts of the same paw. Duty Cycle (%) = (stand/step cycle) × 100.

Muscle strength was determined using a gripmeter (ALG01; Bioseb, Vitrolles, France). The mouse was placed over a metallic grid that it instinctively grab to try to stop the involuntary backward movement carried out by the manipulator until the pulling force overcomes their grip strength. After the animal loses its grip, the strength-meter scores the peak pull force. The muscle force (in g) was measured three times per mouse. Results are the mean of three consecutive assays.

All recordings were performed by the same blinded investigator to minimize variability in the experimental procedure.

### 2.4. Electrophysiology

Mice at 6, 12, and 18–24 months of age were anesthetized with a solution of 80 mg/kg ketamine chlorhydrate (Imalgene 1000; Merial, Lyon, France) and 10 mg/kg xylazine (Rompun 2%; Bayer HealthCare SAS, Loos, France) injected intraperitoneally. During the tests, mouse body temperature was kept constant between 34 and 36 °C by means of a thermostat-controlled heating pad. Neurophysiological and electromyography (EMG) recordings were made with a standard electroneuromyograph apparatus (AlpineBiomed ApS, Skovlunde, Denmark) in accordance with the guidelines of the American Association of Electrodiagnostic Medicine. The low-pass filter was set at 20 Hz, and the high-pass set at 10 kHz. These settings were used for all measurements.

EMG was performed as previously described [37,38]. Electrical activity was recorded using a monopolar needle electrode (diameter 0.3 mm; 9013R0312; Medtronic, Minneapolis, MN, USA) inserted into the tail of the mouse (grounding electrode). Recordings were made with a concentric needle electrode (diameter 0.3 mm; 9013S0011; Medtronic). Spontaneous electrical activity was monitored in *gastrocnemius* muscle on both legs for at least 2 min. Only spontaneous activity with peak-to-peak amplitude of at least 50 µV was considered to be significant.

Compound muscle action potentials (CMAP) were recorded in *gastrocnemius* muscle as described previously [39]. Briefly, CMAPs were elicited with square pulses of 0.2 ms duration with a repetition rate set at 1 Hz, delivered with a monopolar needle electrode to the sciatic nerve at the sciatic notch level. CMAPs were measured by two monopolar needle electrodes inserted in the gastrocnemius muscle, and the system was grounded by subcutaneously inserted monopolar needle electrode in the back of the animal. Data were acquired with a sensitivity between 1 and 10 mV/division with a sweep speed of 1–2 ms/division to ensure accurate measurement of low-amplitude responses. Supramaximal responses were gradually generated, and maximal responses were obtained with stimulus currents <5 mA. Amplitudes (mV) from the baseline to the maximal negative peak of the CMAPs were measured and averaged, resulting in one average CMAP amplitude per animal, which was used for statistical analysis. 

Repetitive nerve stimulations (RNS) were performed with an intensity producing supramaximal and stable sciatic nerve excitation, resulting in stable CMAPs in the *gastrocnemius* at low frequency stimulation. The stimulus level remained supramaximal throughout the test. Trains of 10 stimulations were applied at 20, 50, and 100 Hz. For each frequency, three trains of stimulations separated by a 15 s break were delivered, and between each frequency a 2.5 min break was performed to allow complete recovery. The amplitude change was calculated according to the following formula: (amplitude of the 1st CMAP—amplitude of the 5th or the 10th CMAP)/amplitude of the 1st CMAP and expressed as %. If comparison of the tenth evoked action potential with the first shows a decrement of ≥10% in amplitude, there is a significant positive decremental response to supramaximal stimulation.

### 2.5. Tissues Preparation

For neuromuscular junction analysis, mice were deeply anesthetized with 120 mg/kg ketamine, 16 mg/kg xylazine injected intraperitoneally, and perfused with 4% ice-cold paraformaldehyde in 0.1 M phosphate buffer pH 7.4. Tissues were post-fixed in the same fixative for 24 h and kept at 4 °C in PBS containing 0.002% thimerosal until use.

For biochemistry and histochemistry, mice were deeply anesthetized with 120 mg/kg ketamine, 16 mg/kg xylazine injected intraperitoneally, and killed by decapitation. Muscles were either snap frozen in liquid nitrogen (biochemistry) or embedded in Tissue-Tek O.C.T. compound (Sakura Finetek France SAS, Villeneuve d’Ascq, France) and snap frozen in isopentane (histochemistry) and stored at −80 °C until used. 

### 2.6. Neuromuscular Junctions

Bundles of muscles fibers were prepared from *extensor digitorum longus* (EDL) muscle under a binocular loupe, and labelled with antibodies directed against synaptophysin (1/100, Eurogentec N°1412429: H-CAP PGA PEK QPA PGDA-NH2, Eurogentec, Seraing, Belgium) and with rhodamine-conjugated alpha-bungarotoxin (αBGT, 1 mg/mL; Sigma-Aldrich Chimie Sarl, saint quentin Fallavier, France) and Hoechst 33342 (1 mg/mL; Sigma-Aldrich Chimie Sarl, saint quentin Fallavier, France) using a standard protocol. YFP-EDL muscles were only incubated with rhodamine-conjugated αBGT and Hoechst 33342. 

Immunofluorescence stainings were monitored with laser scanning microscope (confocal LSM 800 Zeiss, Carl Zeiss Microscopy GmbH, Jena, Germany) equipped with 20× objective for synaptophysin experiment, or 40× oil objective for YFP^+^ experiments. Excitation rays were sequential diode 405 nm, argon laser 488 nm, diode 561 nm. Emission bandwidths were 400–500 nm for Hoechst 33342, 520–570 nm for Alexa488 and 570–617 nm for rhodamine. Z-Stack images (1.5 µm optical section, 20 frames per stack) were acquired and each stack of 30 µm was aplaned and analyzed using ImageJ freeware (http://imagej.nih.gov; accessed on 20 March 2022). All analysis were conducted by an observer blind to the animal’s group membership. To assess the area occupied by either αBGT or synaptophysin staining, images exported in ImageJ were thresholded. Two thresholds were used for synaptophysin: a low threshold adjusted to best fit the whole synaptophysin staining and a high threshold fiting with the high intensity of the synaptophysin staining. Such measures were used to calculate the percentage of synaptophysin staining covering the endplate labelled with αBGT measured with a single threshold.

### 2.7. Electron Microscopy

Mice from both sexes were used for electron microscopy. Mice were anesthetized by intraperitoneal injection of 100 mg/kg ketamine chlorhydrate and 5 mg/kg xylazine and transcardially perfused with 2.5% glutaraldehyde and 2.5% paraformaldehyde in cacodylate buffer (0.1 M, pH 7.4). EDL were dissected and immersed in the same fixative overnight. After three rinses in cacodylate buffer (EMS, 11650), samples were post-fixed in 1% osmium tetroxide in 0.1M cacodylate buffer for 1 h at 4 °C and dehydrated through graded alcohol (50, 70, 90, and 100%) and propylene oxide for 30 min each. Samples were oriented and embedded in Epon 812. Semithin sections were cut at 2 µm with an ultramicrotome (Leica Ultracut UCT, Leica Biosystem, GmbH, Nussloch, Germany), stained with 1% Toluidine blue in 1% sodium borate and examined by Leica optical microscope (LEICA DMLB, Leica Microsystems GmbH; Mannheim, Germany). Ultrathin sections were cut at 70 nm, contrasted with uranyl acetate and lead citrate, and examined at 70 kv with a Morgagni 268D electron microscope (FEI Electron Optics, Eindhoven, The Netherlands). Images were captured digitally by Mega View III camera and Soft Imaging System (Olympus Soft Imaging Solutions, Munster, Germany).

### 2.8. Histochemistry

Serial 14µm-thick cross-sections of TA were cut with a cryostat at −20 °C, mounted on slides and stored at −20 °C until use. For SDH staining, sections were fixed for 15 min at 4 °C in acetone and air dried. They were incubated for 30 min at 37 °C in 0.2M Tris buffer pH 7.4 containing 2 mg/mL nitroblue tetrazolium, 10 mg/mL sodium succinate, and 25 µg/mL mg phenazine methosulfate. Sections were rinsed in distilled water and mounted in aquapolymount. Microphotographs were obtained with an Eclipse E800 microscope (Nikon, Nikon Metrology SARL, Lisses, France). Quantification of SDH fibers was determined from digitized muscle sections using ImageJ software. For each mouse, cross-sectional area of 100 SDH positive and 100 SDH negative muscle fibers from three areas per section were measured. 

Myofibrillary actomyosin ATPase activity staining was performed as previously described by Pestronk [40], with slight modifications. Briefly, sections were air-dried at room temperature and incubated for 5 min in preincubation solution at pH 4.53 or 4.31 (sodium barbital 37.5 mM, sodium acetate 37.5 mM, adjusted to pH 4.53 or 4.31 with HCl). The sections were then washed in distilled water and incubated at 37 °C in a solution containing ATP (3.6 mM ATP, 26 mM sodium barbital, 18 mM CaCl_2_, pH 9.4) for 25 min. Sections were washed in distilled water, rinsed for 10 min in a 1% (*w*/*v*) CaCl_2_ solution, and 10 min in 2% (*w*/*v*) CoCl_2_ solution, rinsed 4 times in distilled water and incubated 3 min in a 1% (*v*/*v*) ammonium sulfide solution. After being washed with water, sections were dehydrated with ascending concentrations of alcohol and mounted in mounting medium. Muscle fibers were labeled with respect to the 4 major types of fibers (1, 2A, 2B, and 2X) on the basis of differences in the staining intensity for ATPase after acid preincubation [41]. According to the staining intensities, the following classification was used: pH 4.31 (type 1: darkest, type 2X: intermediate and type 2A and 2B: lightest); pH 4.53 (type 1: darkest, type 2A: lightest, and type 2B: grey and type 2X: dark grey).

### 2.9. RNA Extraction and Real-Time Quantitative Polymerase Chain Reaction

Total RNA was prepared following standard protocols. Briefly, each frozen sample (TA) was placed into a tube containing a 5-mm stainless steel bead. Working on ice, 1 mL Trizol reagent (Invitrogen, Groningen, The Netherlands) was added, and homogenization was performed twice in a TissueLyser (Qiagen, Les Ulis, France) at 30 Hz for 3 mins. Samples for RNA-sequencing analysis were then processed using the RNeasy Mini Kit (Qiagen, 74104, Qiagen, Les Ulis, France) according to manufacturer instructions. For RT-qPCR, RNA was extracted with chloroform/isopropyl alcohol/ethanol technic. Samples were stored at −80 °C until use.

One microgram of total RNA was used to synthesize cDNA using Iscript reverse transcriptase (iscriptTM Reverse Transcription Supermix for RT-qPCR, Bio-Rad, Marnes-La-Coquette, France) as specified by the manufacturer. Quantitative PCR was performed on a CFX96 Real-time System (Bio-Rad, Marnes-La-Coquette, France) using iQ SYBR Green supermix (Bio-Rad, Marnes-La-Coquette, France). PCR was performed as follow: 95 °C for 30 s, followed by 40 cycles of 4 s at 95 °C and 4 s at 60 °C. Three standard genes (*H1H2BC* and *H2AC* histones, and Tata-box binding protein [*Tbp*]) were used and data were normalized with GeNorm softwarev3.5. Primer sequences are given in Appendix A.

### 2.10. Statistical Analysis

Unless otherwise indicated, data are expressed as the mean ± SEM. GraphPad Prism version 6.0 a software was used for statistical analysis. Tests used are indicated in the legends under the figures. Differences with *p*-values < 0.05 were considered significant.

## 3. Results

### 3.1. Neuronal Expression of the CHMP2B^intron5^ Mutant Is Sufficient to Trigger Alterations in Gait and Motor Coordination

To determine to which extend neuron-specific expression of CHMP2B^intron5^ could impact the motor performance of the transgenic mice, the motor skills of HE and nTg mice were assessed using the catwalk gait analysis system. Mice were studied at 6, 12, and 18–24 months of age initially defined as asymptomatic, early symptomatic and symptomatic stages [32] and 3 parameters were measured, the hind paws spacing (base of support), the regularity index (percentage of normal step sequences) which assesses coordination between limbs, and the duty cycle corresponding to the stance duration during a step cycle (Figure 1A). 

In HE mice, hind paws spacing corresponding to the base of support was significantly increased by 8.5% at 6 months, by 14% at 12 months, and by 20% at 18 months of age compared to age-matched nTg mice. In contrast, front paws spacing was unaltered in HE mice compared to nTg (Appendix A). Normal at 6 and 12 months, the regularity index was significantly decreased by around 12% at 18–24 months for HE mice compared to nTg mice. We also assessed the duty cycle—the average percentage of time (in a stride) that a mouse paw is in contact with the glass walkway—for all four paws. The duty cycles of the right (Figure 1A) and the left (Appendix A) front paws were significantly increased from 12 months of age in HE mice when compared with nTg mice. However, the changes in duty cycle were less pronounced for the hind paws, appearing later at 18–24 months of age, with a significant decrease of 15% for the right paws and a decreasing trend for the left paws for HE compared to nTg mice (Appendix A). Taken together, these data show that neuronal expression of CHMP2B^intron5^ leads to impaired posture and gait associated with decreased motor coordination that progress with age in HE mice. 

Having shown these changes, we evaluated the grip strength of these mice and found a decrease of around 21% in muscle strength of HE compared to nTg mice at the symptomatic age of 18–24 months (Figure 1B). At the same age, the body weight was found to be reduced by 15% (Figure 1C) and the weight of *tibialis anterior* (TA), *gastrocnemius* and *soleus*, 3 hindlimb muscles was found to be reduced by 30–40% (Figure 1D and Appendix A). At the molecular level (Figure 2), mRNA levels of *atrogin-1*—a gene involved in muscle atrophy [42]—were 2.5-fold higher at 18–24 months. mRNA levels of *AChR*α—whose expression is induced with denervation or synaptic dysfunction [43]—were progressively increased from 6 months of age reaching a 4-fold induction in HE mice compared to nTg mice, while mRNA levels of *AChR**ε* and *Musk* were only increased at 18–24 months by ~3-folds, and AChRγ mRNA levels were decreased by 30%. 

### 3.2. Expression of CHMP2^intron5^ Results in Structural and Functional Alterations of the Neuromuscular Junction

Having shown here that alteration of the motor functions and reduced muscle strength were associated with muscle atrophy at 18–24 months, we wondered to which extend NMJs were morphologically altered at this age. To address this question, we first looked for morphological evidence of NMJ denervation based on whether the post-synaptic end-plate labelled with fluorescent αBGT lacked, either fully or partially, an overlying nerve terminal visualized with YFP. We found that in both genotypes, more than 90% of the NMJ were contacted by a YFP^+^ terminal. However, YFP fluorescence did not systematically covered the post-synaptic endplate in HE suggesting partial denervation by localized retractation of the nerve terminal (Figure 3). 

We next examined the distribution of synaptic vesicles in the presynaptic nerve terminals using an anti-synaptophysin antibody and its overlaping with the post-synaptic endplate labelled with rhodamine-αBGT. As shown in Figure 4A, synaptophysin immunoreactivity was diffusely distributed in the pre-synaptic terminals of nTg mice, covering the post-synaptic endplate between the crests of the strongly αBGT-labeled junctional folds. In contrast, nerve terminals of HE mice exhibited a patch-like synaptophysin immunoreactivity that did not fully cover the post-synaptic endplate as evidenced by the lack of systematic colocalization between synaptophysin and αBGT labelings.

NMJs were then analyzed individually to calculate the overlap of synaptophysin immunoreactivity with the corresponding endplate visualized with αBGT. No significant change was found for the post-synaptic endplate area in HE mice compared to nTg mice (Figure 4B). However, areas covered by both total and high intensity synaptophysin stainings were reduced (Figure 4C). In addition, the ratio high intensity synaptophysin area/αBGT area was reduced in HE compared to nTg mice although not different for the total synaptophysin (Figure 4D). Nevertheless, the distribution within the population of NMJ was changed. In 75% of nTg NMJs, total synaptophysin covered more than 60% of the postsynaptic end-plate whereas only 50% of HE NMJs showed the same overlap (Figure 4E). At the opposite, high intensity synaptophysin covered less than 40% of the post-synaptic endplate in 69% of nTg NMJ and this value was increased to 96% in HE (Figure 4F). 

Altogether these data show a reduction of synaptophysin staining with an abnormal distribution, suggesting a decrease of synaptic vesicles within the presynaptic terminals.

In electron microscopy (Figure 5), NMJ of HE mice were also strikingly altered when compared to NMJ of nTg mice. Indeed, in most of the pre-synaptic terminals, the amount of synaptic vesicles was decreased. In addition, large electron-dense deposits, autophagosomes, and enlarged autolysosomes were present in the terminals, along with an apparent increase in terminals volume. At the end-plate level, a marked reduction of the post-synaptic folds was noticeable in many NMJ.

To assess the functionality of the NMJ, we then performed in vivo electrophysiological recordings. We first used electromyography (EMG) to record electrical activity in skeletal muscle of anesthetized mice. Recording of the resting activity in the *gastrocnemius* muscle of 18–24-month-old HE mice consistently revealed abnormal spontaneous activity with few fibrillation potentials and in some rare cases fasciculations (Figure 6A). This activity was rarely detected in the control mice of the same age and was absent in younger HE mice (not shown). We then measured the amplitude of the compound motor action potential (CMAP) in the *gastrocnemius* muscle obtained by a single supramaximal stimulation of the sciatic nerve, and tested the neuro-muscular transmission by performing repetitive nerve stimulations (RNS). CMAP amplitudes were comparable between genotypes at the three stages tested (Figure 6B and Appendix A). 

At 6 and 12 months of age, RNS did not significantly alter CMAP amplitudes at the tested frequencies, regardless of genotype (Appendix A). However, at 18–24 months, RNS at 20, 50, and 100 Hz induced a decremental response in HE mice, which increased as the frequency was ramped up. At 20 Hz, 4 HE mice out of 14 (28%) had a decrement greater than or equal to 15%, this number of mice increased to 6 out of 14 (42%) at 50 Hz and to 9 out of 12 (75%) at 100 Hz (Figure 6C–E). In contrast, CMAP amplitude tended to increase in nTg mice at 100 Hz.

These data suggest that MNs expressing CHMP2B^intron5^ cannot sustain reliable neurotransmission at the NMJs and exhibit abnormal signs of synaptic depression at 18–24 months of age.

### 3.3. Neuronal Expression of CHMP2B^intron5^ Leads to Impaired Muscle Metabolism and Myosin Composition

Adaptive changes of muscle fibers can occur in response to variations in the pattern of neural stimulation. Indeed, nerve activity influences the contractile and metabolic properties of muscle fibers [44]. We thus asked whether neuronal expression of CHMP2B^intron5^ induced muscle metabolic reprogramming associated with change of myosin heavy chain (MyHC) types as described in SOD1 ALS mice. We first assessed SDH enzymatic activity by histochemistry in TA. No obvious difference in SDH staining was seen between genotypes until 18–24 months of age (Figure 7). However, at 18–24 months of age, overall SDH staining was increased in HE mice compared to nTg mice. The staining became more heterogenous within SDH^+^ fiber population, with many fibers being slightly purple while others were deep purple. Analysis of the cross-sectional area (CSA) frequency distribution of SDH^+^ and SDH^−^ muscle fibers showed a shift toward small-caliber SDH^+^ and SDH^−^ muscle fibers in TA of HE mice compared to nTg mice at 6 and 18–24 months of age (6 months: SDH^+^ fibers: interaction CSA × genotype *p* = 0.025; SDH^−^ fibers interaction CSA × genotype *p* = 0.0026; 18–24 months: SDH^+^ and SDH^−^ fibers: interaction CSA × genotype *p* < 0.0001). This shift to small CSA was not observed at 12 months.

To more precisely delineate the metabolic profile of TA muscle fibers, we quantified the expression of biochemical markers associated with muscle contractile properties and oxidative or glycolytic metabolism by RT-qPCR. We first analyzed the mRNA levels of four *MyHC*: *MyHC 4*, expressed in type 2B glycolytic fast-twitch fibers, *MyHC 1* and *MyHC 2* found respectively in mix type 2X and 2A glycolytic/oxidative fast-twitch fibers and *MyHC7* expressed in type 1 oxidative slow-twitch fibers (Figure 8A). 

At 6 months of age, mRNAs of *MyHC 1* and *MyHC 4* were increased by approximately 2-folds in HE mice compared to nTg mice. *MyHC 1* remained elevated at 12 months and returned to nTg level at 18–24 months whereas *MyHC 4* decreased to nTg level by 12 months. In parallel, *MyHC 2* was doubled from the age of 12 months and *MyHC7* was increased by 2.5 times at 18–24 months although not reaching significance. We also measured molecular markers of fast twitch fibers (*RyR1*: ryanodine receptor 1, *Parv*: parvalbumine, *Tnnc2*: troponin C2) and slow-twitch fibers (*TnnC1*: troponin C1). *RyR1* and *Parv* expression levels were ~4-fold increased at 6 months in HE mice compared to nTg mice and returned to basal levels from 12 months. *TnnC1* and *TnnC2* expressions were comparable between genotypes at all ages. Finally, we assessed the expression of genes involved in mitochondrial activity and oxidative metabolism (Figure 8B,C). As shown in Figure 8B, expression of *Tfam* (a mitochondrial transcription factor acting as a key activator of mitochondrial transcription) was similar in the two genotypes, irrespective of the age. In contrast, mRNA levels of mitochondrial respiratory chain components *Sdh*α (complex II: major catalytic subunit of succinate-ubiquinone oxidoreductase), *Uqcrc2* (complex III: complex III Cytochrome B-C1 Complex Subunit 2), *Cox5A* (Complex IV: subunit of Cytochrome c oxidase, the terminal enzyme of the mitochondrial respiratory chain) were progressively increased to reach significance at 18–24 months with a 2- to 3-fold increase. In parallel, mRNA levels of *Atp5g3* (a subunit of mitochondrial ATP synthase) remained unchanged (Figure 8C). The expression of *Ppar**δ* (peroxisome proliferator-activated receptor δ) was significantly increased by about 3-fold at 6 months in HE mice compared to nTg mice. Its expression remained stable at 12 months before decreasing to the level of nTg mice at 18–24 months. Surprisingly, *Pgc1*α (peroxisome proliferator activated receptor gamma co-activator 1-alpha), the master regulator of mitochondrial biogenesis and function was only transiently induced by around 2 times at 12 months of age while *Pppar*α (peroxisome proliferator-activated receptor α) and *Pdk4* (pyruvate dehydrogenase kinase 4) were increased at 12 and 24 months in HE mice compared to nTg mice. Finally, *myoglobin* expression was progressively increased with age in HE mice compared to nTg mice although without reaching significance. Altogether, these data showed a progressive switch of metabolic capacity of the TA muscle from glycolytic to oxidative properties associated a change in myosin heavy chain from fast-twitch to slow-twitch fibers. 

To confirm the switch of muscle fiber type we performed a myofibrillar actomyosin ATPase staining on TA serial sections to compare ATPase activities within the same myofibers. By changing the pH of preincubation buffer, this technique allows to discriminate the fourth types of muscle fibers. At pH 4.31, type 1 fibers are the darkest, type 2X exhibit intermediate staining, and type 2A and 2B are the lightest/colorless fibers. At pH 4.53, type 1 fibers are the darkest, type 2A are the lightest (colorless), type 2B are brown/grey, and type 2X are dark brown/grey. For each age and genotype, small magnifications of representative stainings are shown in Appendix A (6 and 12 months) and Figure 10 (18–24 months).

At 6 months of age (Figure 9A), no difference was found between genotypes. Three categories of fibers were identified. Fibers dark at pH 4.31 and pH 4.53 corresponding to type 2X fibers: they represented around 30% of the population. Fibers colorless at pH 4.31 and brown/grey at pH 4.53 corresponding to type 2B fibers: they represented around 60% of the population, and a third type of fibers, colorless at pH 4.31 and brown/grey at pH 4.53 corresponding to a transition fiber type which represented around 10% of the population. No type 1 or type 2A fibers was found. At 12 months of age (Figure 9B) in HE mice the proportion of transition fibers was increased reaching around 23% and the proportion of type 2X and 2B was decreased to respectively 24% and 53%. At this age the proportion of fiber types remained unchanged in nTg mice. At 18–24 months (Figure 10), the myofibrillar ATPase staining pattern became highly altered in HE mice compared to 12 months of age (Figure 9B) and to age-matched nTg mice (Figure 10). At pH 4.31 the staining of type 2X was strongly decrease compared to 12 months. At pH 4.53 transition fibers became lighter and their number was increased. The difference in staining intensity between brown and dark brown fibers was hardly distinguishable, making a quantification of fiber types impossible at this age. A few rare black fibers corresponding to type 1 fibers were also detected at this age (Figure 10 inset). 

Altogether, these data support a switch of metabolic and contractile properties of fast glycolytic muscle as disease progresses.

## 4. Discussion

In Human, CHMP2B mutations induce either ALS or FTD or both diseases. Previous in vitro and in vivo studies have shown that CHMP2B^intron5^ has a profound impact on neuronal homeostasis and alters post-synaptic structure, causing severe dendritic retraction with impaired morphological maturation of dendritic spines [45,46,47,48]. Although expressed at a lower level, CHMP2B is also present in axon terminals [48]. Previously, CHMP2B^intron5^ has been involved in autophagic and endo-lysosomal pathways dysfunctions (for review see [49]). However, its roles in the presynaptic compartment remains elusive. Transgenic mice expressing CHMP2B^intron5^ mutant in neurons develop a late onset disease characterized by a progressive axonal axonopathy in association with decreased grip strength and gait deficits [31,32,33], suggesting a possible deleterious effect of this mutation in the presynaptic compartment of the NMJ. Of note, transgenic mice expressing wild-type CHMP2B do not exhibit any motor or behavioral phenotype [31]. As alterations in synaptic transmission at the NMJ are some of the earliest changes observed in ALS mouse models and patients [3,4,5,50] we therefore examined the consequences of neuronal expression of the CHMP2B^intron5^ mutant on NMJ functionality in vivo, and on skeletal muscle properties. To this end, we used a transgenic line in which CHMP2B^intron5^ expression was driven by the Thy1.2 promoter [32] and was, therefore, restricted to neurons [51].

In HE mice, subtle changes of gait were already detectable at 6 months of age. These alterations of gait precede a detectable decrease of grip strength already reported [32] and muscle mass characteristics of sarcopenia that are only detectable at 18–24 months. A decrease in muscle mass together with the increase in *atrogin-1* expression, as shown in this study, was also reported in skeletal muscles of ALS patients and ALS mouse models [11,14,52]. Atrophy of muscle fibers attested in this study by the decrease of CSA of SDH^+^ and SDH^−^ myofibers could reflect either effective denervation [53,54] or synaptic dysfunction of the NMJ [55]. In our CHMP2B^intron5^ HE mice, the presence of YFP^+^ nerve terminals in contact with the post-synaptic endplates showed the physical connection between nerve terminals and endplates. However, in agreement with previously published results by us and others [31,32], the absence of total overlap of the endplate together with the irregularity of synaptophysin staining support the notion of a partial denervation. 

In line with these results, the limited increase in nicotinic *AChR*α and *AChR*ε subunits expression without induction of *AChR*γ, associated with the late increase in *Musk* expression (responsible for the characteristic clustering of nicotinic AChRs and essential for the maintenance of NMJ) may reflect a long-term synaptic dysfunction or a partial denervation rather than massive denervation. Indeed, after a nerve transection leading to chronic denervation or a pharmacological blockade of NMJ, the pattern of nicotinic AChR expression evolves from an adult form toward a fetal form with re-expression of *AChR*γ, prior to returning to the adult form with a re-expression of *AChR*ε [42,56,57]. The rather modest spontaneous activity of denervation detected by EMG and our results obtained with the RNS experiments also support the hypothesis of a deficit in synaptic activity rather than a major denervation. Indeed, in other mouse models of ALS, up-regulation of *AChR*α reaches a factor of 20-fold. At that time, more than 60% of MN are lost, EMG records high levels of fasciculation and full interferential pattern of spontaneous electrical activity, and mice are paralyzed [11,58].

In clinical practice, RNS is used for the evaluation of patients with suspected neuromuscular transmission disorders. Under physiological condition, CMAP is stable or can increase in response to isolated or repetitive supramaximal stimulations respectively. Indeed, in response to electrical stimulation, nerve terminals are depolarized and release a quanta of acetylcholine (Ach) into the synaptic cleft by fusion of the synaptic vesicles with the presynaptic membrane. ACh binds to nicotinic AChRs on the post-synaptic end-plate, causing an action potential that induces muscle contraction. To maintain efficient neurotransmission, synaptic vesicles undergo repeated cycles of exocytosis, endocytosis, and vesicles formation. The rate of the turn-over is high enough to ensure a safety factor that will guaranty sustain neurotransmitter release and support prolonged high frequency stimulation without changing the muscle response.

Decremental response to RNS has been reported in ALS patients [59,60,61]. In this study, at 18–24 months of age, after a single supramaximal stimulation, CMAP amplitude is comparable in HE and nTg mice, showing a sufficient and efficient ACh release when stimulation is isolated. In normal subjects, at high frequency, an increase of CMAP amplitude can be observed due to the facilitation enhanced by sustained presynaptic calcium influx, which is the case at 100 Hz in nTg mice. However, CMAP amplitude is decreased when HE mice are subjected to RNS at high frequency. Similar reduction of motor response amplitude was also reported after high frequency stimulation in pre-symptomatic SOD1^G93A^ mice and in FUS^P525L^ mutant mice [62,63]. The abnormal fall in CMAP amplitude seen in HE mice suggests a drop in safety factor for transmission originating from the pre-synaptic terminals. In line with this hypothesis, we found a change in the distribution of synaptophysin, an integral membrane protein of the synaptic vesicles, that become clustered in some area of the nerve terminals and absent in other area instead of being homogenously distributed throughout the terminals. Similar observations have been made in SOD1^G93A^ mice [64], in hFUS mice [65], and in the other transgenic line expressing CHMP2B^intron5^ [31]. Consistent with this change of synaptophysin immunoreactivity, we observed a global decrease of synaptic vesicle in electron microscopy which could explain the alteration of the synaptic transmission found at the NMJ. Congruent to these data, a very recent and elegant in vitro study by Clayton and collaborators [66] demonstrated that the presence of CHMP2B^intron5^ mutant induces a defect of synaptic vesicles trafficking at the presynaptic terminal of neuro-neuronal synapses leading to a defective synaptic vesicles exocytosis upon a 40 Hz stimulation. They also showed a decreased number of synaptic vesicles belonging to the recycling pool (a pool of vesicles recycled upon moderate stimulation and supplied under physiological stimulation) as well as the reserve pool which is mobilized in response to high frequency stimulation [67]. Decreased neurotransmitter release in presymptomatic SOD1^G37R^ mutant mice, and quantum release and synaptic transmission attenuation were also observed in TDP-43^Q331K^ and FUS mutant mice, reinforcing the role of synaptic transmission defects in ALS pathology [65,68,69].

In accordance with our own work reported here, Clayton and collaborators point toward an accumulation of endosomes, a phenomenon that has already been documented in a number of CHMP2B cellular and animal models [21,31,44,70,71,72,73] and in patients [74]. This increase in endosomes may reflect a reduction of the endolysosomal pathway efficiency due to reduced fusion of endosomes with lysosomes, and an alteration of the autophagic pathway [74], two processes that may participate in the dysfunction of the presynaptic transmission. Altered autophagy has also been reported in other mouse models of ALS (for review see Amin [75]). The accumulation of endosomes, autophagosomes and electron-dense structures may also participate in the apparent increase in presynaptic terminal volume observed electron microscopy in HE mice.

Taken together, these results show that neuronal expression of CHMP2B^intron5^ induces morphological changes in the presynaptic element and leads to abnormalities in neurotransmission that precede motor axon withdrawal. 

More importantly, our results also show that these changes modify the muscle counterpart of the NMJ. Four major types of fibers are found in skeletal muscles of mammals. Each fiber type is defined by the presence of a specific isoform of MyHC, by a distinct program of gene expression and specific metabolism [44]. Type 1 or slow-twitch fiber expresses MyHC 7 and is characterized by low speed of shortening, high resistance to fatigue, high level of myoglobin, and oxidative metabolism. This fiber type is found in slow-twitch fatigue-resistant motor units. Type 2 fibers that are fast-twitch include three subtypes of fibers: type 2B expressing MyHC 4 and characterized by a glycolytic metabolism, highest speed of contraction, and lowest resistance to fatigue; type 2A expressing MyHC 2 and characterized by oxidative metabolism, lowest speed of shortening, and the highest resistance to fatigue, and are, therefore, more similar to type 1 fibers; and type 2X expressing MyHC 1, falling between these extremes with mixed glycolytic-oxidative metabolism. Type 2B and 2X fibers are found in fast-twitch, fast fatigable (FF) motor units while type 2A fibers are found in fast-twitch, fatigue-resistant motor units. Thus, the metabolism of a muscle fiber is tightly linked with its contractile properties. Muscle fibers are dynamic structures capable of changing their phenotype under various conditions including altered neuromuscular activity. Following disruption of their nerve supply slow muscles become faster and fast muscles become slower. The changes in MyHC isoforms follow a general scheme of sequential transitions from fast-to-slow and slow-to-fast: Type 2B ↔ 2X ↔ 2A ↔ 1 with hybrid fibers bridging the gaps between the pure fibers [76]. During disease progression, a switch of muscle fiber types from glycolytic to oxidative has been described in ALS patients [9] and in mutant SOD1 ALS mouse models [11,13,14,76,77,78]. At the molecular level, this transformation is accompanied by the simultaneous induction of oxidative myosin heavy chains and the repression of glycolytic myosin heavy chains [14,78]. 

In agreement with what has been reported in human tissue and in SOD1 mice, we show here a progressive change in muscle contractile and metabolic properties at the molecular level. In TA muscle, even before a clear alteration of motor functions occurs, the global over-expression of type 2B and 2X myosins (respectively, *MyhC4* and *MyhC1* mRNA) associated with the increased expression of ryanodine receptor 1 and parvalbumin—two markers of type 2 muscle fibers—strongly suggest that muscle fibers increase their contractile capacities possibly to counteract, as a compensatory mechanism, the synaptic dysfunctions that may already be present but remain undetected by the little sensitive electrophysiological approach used in this study. As alterations of motor functions progress, a gradual replacement of fast-twitch glycolytic fibers 2B by more oxidative fibers is seen. At the cellular level, myofibrillar ATPase staining highlights the progressive change in muscle fiber type with a gradual increase in transitional fibers. This may reflect the transition from 2B → 2X at 12 months of age, and 2X → 2A at the later stage where large clusters of clear/colorless transition fibers are observed. To ascertain this observation, an immunohistochemical approach could be used to precisely quantify the MyHC composition of these transition fibers. This switch in fiber type observed at the cellular level is accompanied by increased expression of metabolic markers known to promote or participate in oxidative metabolism under physiological condition or in an ALS context such as *Pgc1*α, *Ppar*α, *Ppar**δ*, *Pdk4,* or *Lcad* [11,14,18,79], as well as the rise of mRNA encoding important component of the electron transport chain (*Sdha*, *Uqcrc2*, and *Cox5A*) supporting a higher mitochondrial activity. These changes in mitochondrial or metabolic gene expression reflect the metabolic shift in glycolytic muscle characteristic of ALS, from glycolysis-based to a more oxidative metabolism.

The open question is now to understand how this shift occurs in CHMP2B^intron5^ muscle fibers. In ALS, fiber-type switch correlates with the progressive death of fast-fatigable MNs. Loss of fast-fatigable MNs is compensated, at least at initial stages, through reinnervation of orphan fibers by slow fatigue-resistant MNs, a process achieved via compensatory sprouting that leads to muscle fiber-type grouping [80,81]. We have previously shown that around 70% of the NMJ were partially denervated at 24 months [32]. However, the number of MN was unaffected thus ruling out the loss of MN as responsible for the changes observed in CHMP2B^intron5^ mice.

It is now established that chronic electrical stimulation of glycolitic muscle, such as TA or EDL, with slow MN-specific impulse patterns (tonic low frequency) elicits profound changes in muscle phenotype (for reviews, see [76]). Indeed, such a stimulation induces major changes in myosin expression and fibers progressively switch from 2B to 2A type. It further affects all functional elements of the muscle fiber, including other myofibrillar proteins such as the troponin subunits, protein involved in Ca^2+^ homeostasis (i.e., parvalbumin and RyR), enzyme involved in energy metabolism, and myoglobin content [76]. The changes induced by chronic stimulation at low frequency are thus comparable to those described in this study supporting the importance of the decreased synaptic activity in the muscular phenotype found in HE mice. To support this hypothesis, a similar observation was recently reported in a model of spinal and bulbar muscular atrophy (SBMA), a neuromuscular disease characterized by lower motor neurons loss and skeletal muscle atrophy [82]. In this model, mice developed progressive motor dysfunctions in the absence physical denervation of the myofibers. These alterations were associated with a switch of myofiber type from 2B → 2A/2X and from glycolytic-to-oxidative metabolism. Based on these observations they propose that a structural impairment of the motor unit likely causing aberrant communication between MN and muscle, could ultimately contributes to the loss of motor performance. 

Altogether one can hypothesize that in CHMP2B^intron5^ HE mice, the neuronal expression of the mutant leads to an impairment of MN-muscle communication with a decrease of synaptic activity mimicking a switch of fast-fatigable MNs to slow fatigue-resistant MNs. This hypothesis is thus reminiscent with what has been proposed for ALS patients [80,81]. 

To conclude, our study establishes for the first time that expression of CHMP2B^intron5^ restricted to neurons is sufficient to depress the synaptic transmission at the level of NMJ and consequently modify the properties of the muscle fibers leading to changes characteristic of ALS. More investigation is now needed to further understand how these mechanisms could be targeted to develop new therapeutical strategies to prevent this synaptopathy.

## Figures and Tables

**Figure 1 biomolecules-12-00497-f001:**
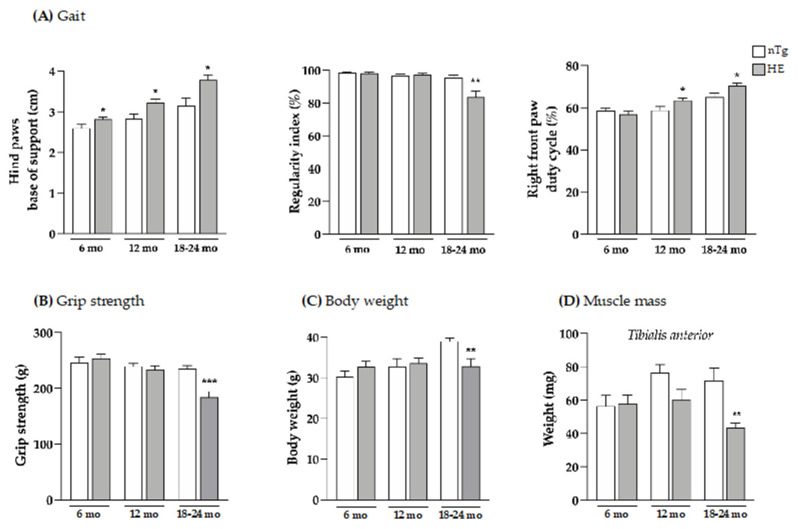
Neuronal expression of CHMP2B^intron5^ induces progressive gait abnormalities and alters muscle parameters. Gait (**A**), grip strength (**B**), body weight (**C**), and weight of *tibialis anterior* muscle (**D**) were evaluated at three ages. HE mice were compared to age-matched nTg mice. Histograms represent means ± SEM; A: n = 9–16/group; B: n = 11–13/group; C: n = 12/group; and D: n = 4–6/group. * *p* < 0.05, ** *p* < 0.01, *** *p* < 0.001, Student’s *t* test (**A**,**B**,**D**) or Mann–Whitney test (**C**).

**Figure 2 biomolecules-12-00497-f002:**
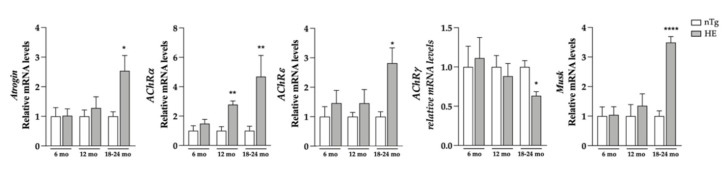
Neuronal expression of CHMP2B^intron5^ induces a modest increase in the expression of denervation markers. Denervation markers were evaluated at three ages. HE mice were compared to age-matched nTg mice. Histograms represent means ± SEM; *Atrogin*: n = 4–6/group. *AchR*α: n = 12/group. *AchR**ε, AchR**γ* and *Musk*: n = 6–8/group * *p* < 0.05, ** *p* < 0.01, **** *p* < 0.0001, Student’s *t*-test.

**Figure 3 biomolecules-12-00497-f003:**
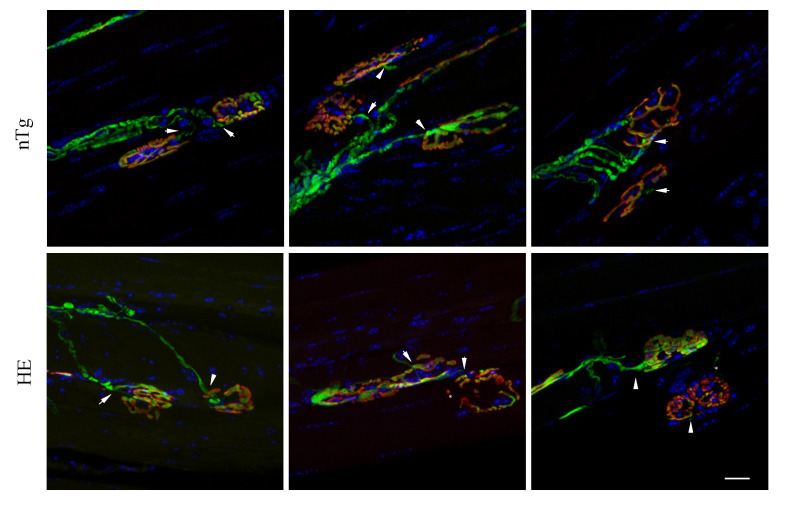
Neuromuscular junctions of HE mice show no evidence of morphological denervation. Representative confocal photographs showing typical EDL neuromuscular junction of 24-month-old HE and nTg mice. Nerve terminals expressing YFP (green) are positioned in close apposition (arrowhead) to the underlying endplate with rhodamine-αBGT-labelled nAChR (red) showing endplates innervation. YFP fluorescence covers endplates of nTg mice unlike HE where the overlap is not systematic (star). Nuclei (blue) are labelled with Hoechst 33342. Scale bar: 20 µm.

**Figure 4 biomolecules-12-00497-f004:**
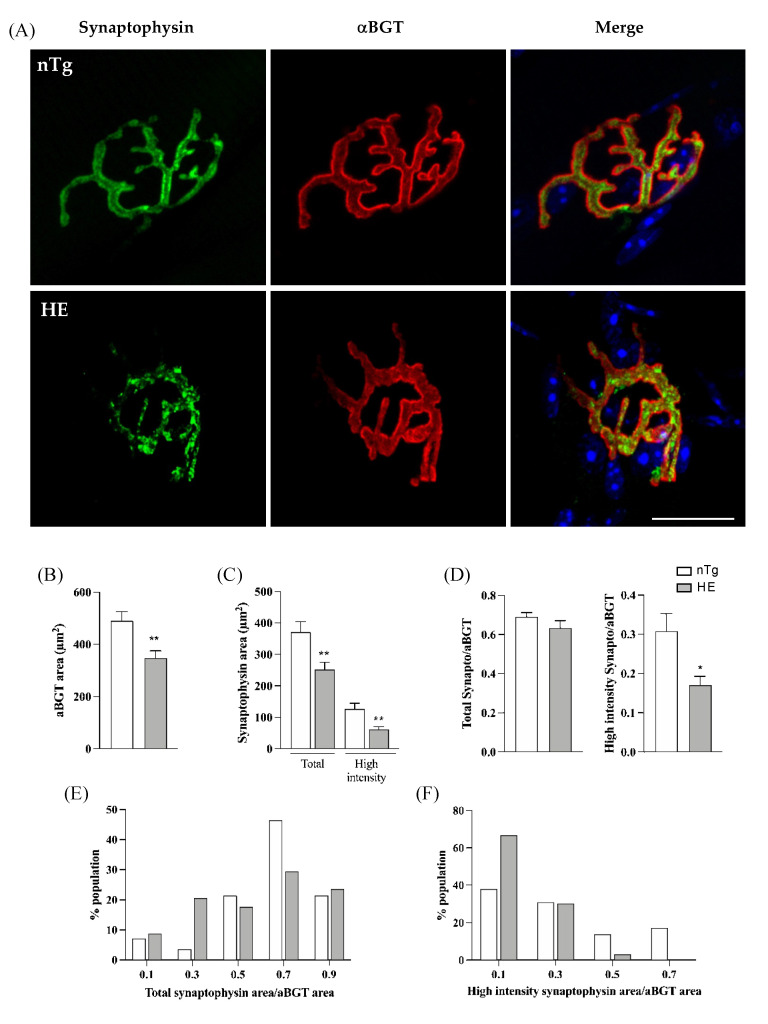
Synaptophysin distribution is altered in NMJ of 24-month-old HE mice. Representative confocal photographs showing typical EDL NMJ of HE and nTg mice (**A**). Rhodamine-αBGT-labelled nAChR delineate motor endplates (red), nerve terminals are labelled by immunodetection of synaptophysin (green) and nuclei (blue) are labelled with Hoechst 33342. Note the dot-like immunoreactivity of synaptophysin in HE mice compared to nTg mice. Scale bar: 20 µm. (**B**) Measure of endplate area by quantification of the area of αBGT staining. No difference is seen between genotypes. (**C**) Measure of area covered by synaptophysin with two thresholds. A low threshold quantifying the whole surface occupied by the green staining corresponding to synaptophysin (Total) and a high threshold quantifying the high intensity of green staining. Note the decrease of synaptophysin area in both condition in HE compared to nTg mice. (**D**) Ratio of synaptophysin area to αBGT for the two thresholds used in (**C**) showing the relative overlap of synaptophysin on endplate; note the significant decrease in the ratio of the high intensity synaptophysin/αBGT in HE compared to nTg mice. (**E**) Distribution of the total synaptophysin/αBGT ratio in the NMJs population analyzed in (**D**). Note the increase in the percentage of NMJ with an overlap ranging from 0.2 to 0.4 and the decrease in the percentage of NMJ with an overlap ranging from 0.6 to 0.8 in HE compared to nTg mice. (**F**) Distribution of the high intensity synaptophysin/αBGT ratio in the NMJs population analyzed in (**D**). Note the increase in the percentage of NMJ with an overlap ≤0.2 and the decrease in the percentage of NMJ with an overlap >0.4 in HE compared to nTg mice. (**B**–**D**) Histograms represent means ± SEM; n = 8–10/mice per genotype and 6–10 JMN/mice. * *p* < 0.05, ** *p* < 0.01, Student’s *t*-test (**B**) or Mann–Whitney test (**C**,**D**).

**Figure 5 biomolecules-12-00497-f005:**
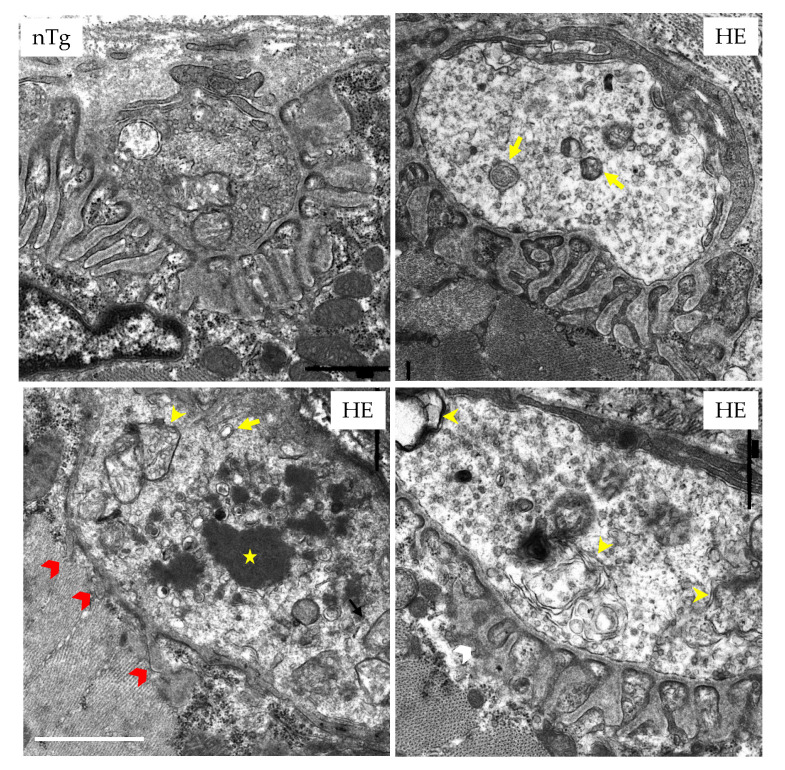
NMJs of HE mice display ultrastructural abnormalities at 18 months of age. EDL were prepared for conventional electron microscopy. Characteristic pictures are shown. In nTg mice, nerve terminal contains numerous small clear synaptic vesicles and mitochondria located in the more proximal portion of the terminal. The postsynaptic membrane has deep junctional folds. In HE mice, the density of synaptic vesicles is decreased in the pre-synaptic terminal that contains enlarged endosomes (yellow arrows), autophagosomes (yellow arrowheads), and electron-dense structures (white star). The post-synaptic folds are occasionally reduced (red arrowheads). Scale bar: 1 µm.

**Figure 6 biomolecules-12-00497-f006:**
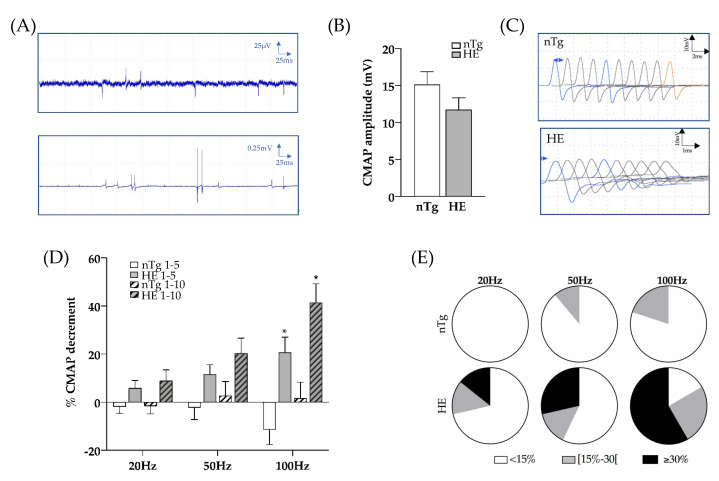
Electrophysiological characteristics of *Gastrocnemius* muscle of CHMP2B^intron5^ HE mice aged 18–24 months. (**A**) Typical EMG recordings obtained in HE mice with fibrillation (top) or fasciculations (bottom). (**B**) CMAP amplitude after a single stimulation of the sciatic nerve at supramaximal intensity. (**C**) Representative CMAP response following RNS stimulation at 100 Hz. (**D**) Quantification of CMAP decrement between the first and fifth stimulations (plain histograms) or the first and the tenth stimulation (dashed histograms) after electrical stimulation of the sciatic nerve at 20, 50 and 100 Hz. (**E**) Proportion of mice with less than 15% decrement (white), between 15 and 30% decrement (grey), and more than 30% decrement (black) in the nTg and HE mice for each frequency of stimulation. Histograms represent means ± SEM; nTg n = 9 HE n = 14. Two ways ANOVA followed by Tukey’s *post hoc* test. * *p* < 0.05, genotype effect: *p* = 0.0008, frequency effect: *p* = 0.0124, and genotype x frequency effect: *p* = 0.0011.

**Figure 7 biomolecules-12-00497-f007:**
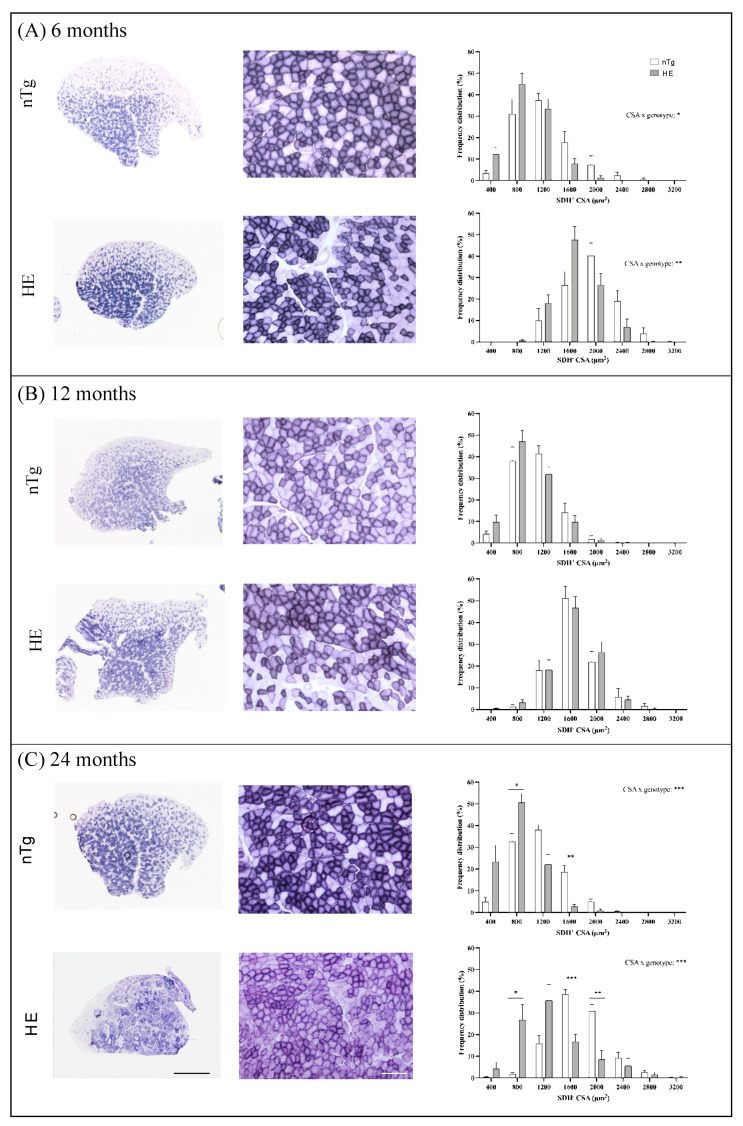
Histochemical detection of SDH activity in TA of nTg and HE mice at 6 (**A**), 12 (**B**), and 18–24 (**C**) months of age. Left: Representative low magnification micrographs showing the global SDH staining in TA. Scale bar: 500 µm. Middle: Representative high magnification micrographs showing the variety of SDH staining within the TA. Scale bar: 50 µm. Right: cross-sectional area (CSA) of SDH^+^ and SDH^−^ fibers were measured and the distribution of the fiber CSA is presented. Note the shift to the left at 6 and 14 months of age for HE compared to nTg mice. Histograms represent means ± SEM, n = 5–10 mice/genotype and 100 SDH^+^ and 100 SDH^−^ fibers were analyzed/mice. Two ways ANOVA followed by Tukey’s *post hoc* test. * *p* < 0.05, ** *p* < 0.01, and *** *p* < 0.001.

**Figure 8 biomolecules-12-00497-f008:**
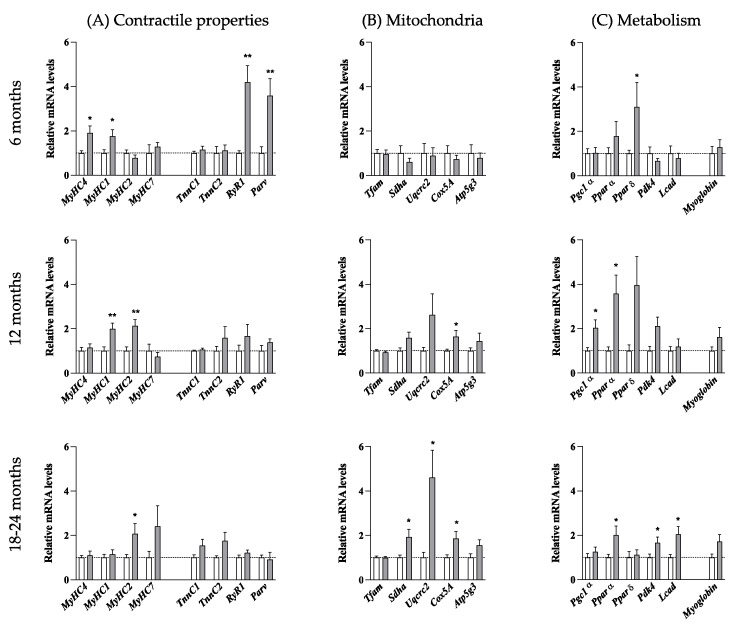
Molecular changes properties of the *tibialis anterior* muscle with age. Relative mRNA levels of molecules involved in contractile properties (**A**), in mitochondria function (**B**), and in metabolism (**C**) were evaluated by qPCR at the indicated ages in tibialis anterior of nTg and HE mice. Histograms represent mean fold change from age-matched nTg mice ± SEM. (**A**): n = 14–20/group, (**B**): n = 6–8/group; (**C**): n = 5–15/group. * *p* < 0.05, ** *p* < 0.01, Student’s *t*-test.

**Figure 9 biomolecules-12-00497-f009:**
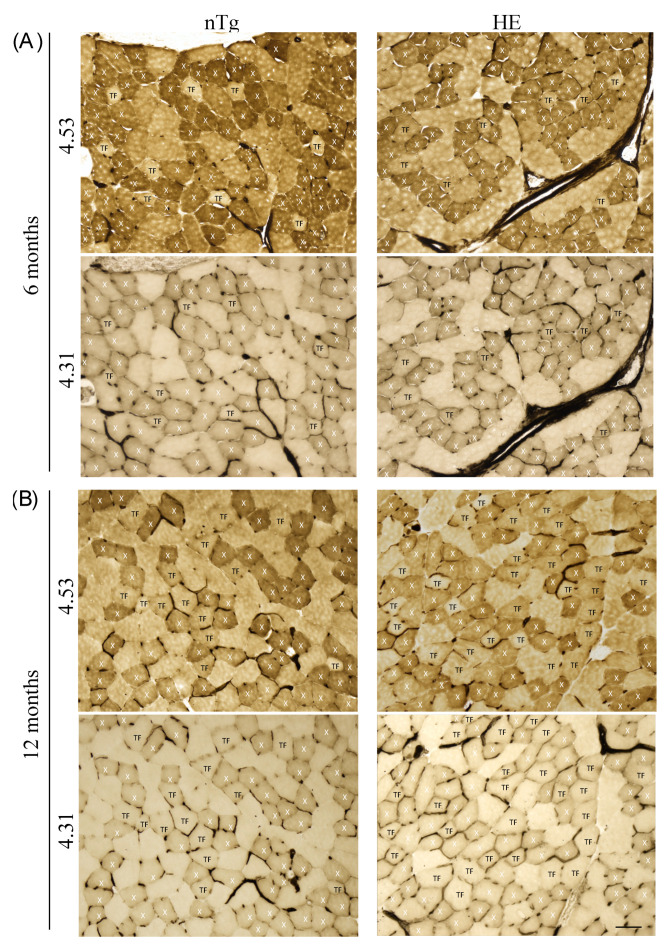
Representative micrographs of serial sections of TA from 6- (**A**) and 12-month-old (**B**) HE and nTg mice after myofibrillar ATPase histochemistry. Serial sections were either preincubated at pH 4.53 or at pH 4.31. At pH 4.53, type 2B fibers are brown, type 2A fibers are the lightest, and type 2X fibers appear dark brown. At pH 4.31, type 2X fibers are light brown and type 2A and 2B fibers are the lightest/colorless. Based on the staining at the two pH, X indicate the 2X fiber type, and TF correspond to transition fibers light brown at pH 4.53 and light/colorless at pH 4.31. Scale bar: 50 µm.

**Figure 10 biomolecules-12-00497-f010:**
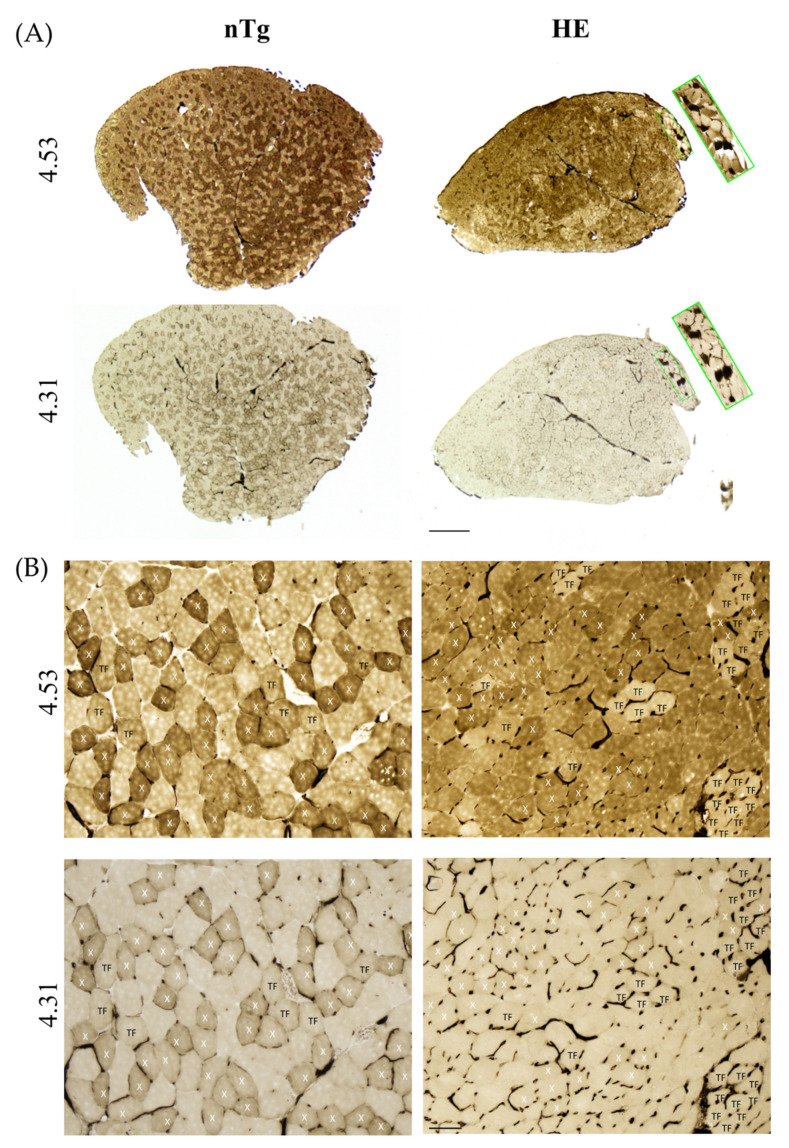
Representative micrographs of serial sections of TA from 24-month-old HE and nTg mice after myofibrillar ATPase histochemistry. Serial sections were either preincubated at pH 4.53 or at pH 4.31. At pH 4.53 and pH 4.31, type 1 fibers are black. At pH 4.53, type 2B fibers are brown, type 2A fibers are the lightest, and type 2X fibers appear dark brown. At pH 4.31, type 2X fibers are light brown and type 2A and 2B fibers are the lightest/colorless. X indicates 2X fiber type, and TF corresponds to transition fibers light brown at pH 4.53 and light/colorless at pH 4.31. (**A**) Low magnification. The inset shows the presence of some type 1 fibers. Scale bar: 500 µm. (**B**) High magnification of the sections presented in (**A**), scale bar: 50 µm. Note the grouping of TFs in HE mice and the more homogeneous brown staining, making it difficult to identify fiber types. Scale bar: 50 µm.

## Data Availability

The data that support the findings of this study are contained in the body of the manuscript or in the Appendix A. Any additional data used are available from the corresponding authors upon reasonable request.

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
