# Peer review of "Alteration of the Neuromuscular Junction and Modifications of Muscle Metabolism in Response to Neuron-Restricted Expression of the CHMP2B^intron5^ Mutant in a Mouse Model of ALS-FTD Syndrome"

_biomolecules, 2022, doi:10.3390/biom12040497_

Round 1
Reviewer 1 Report
Review for Biomolecules Manuscript, biomolecules-1532431
The authors characterized the phenotypes in neuromuscular junction and muscle metabolism of CHMP2Bintron5 mutant mouse, which was a mouse model of ALS-FTD syndrome. In this manuscript, they found (1) the alteration of the neuromuscular junction using behavior tests (Fig.1), immunostaining (Fig.2), electron microscopy (Fig.3), and electroneuromyography (Fig.4). As well, they found (2) the modifications of gene expressions in muscle (Fig.5). Although they claimed that the mutation modified metabolism of muscle in the title, they only showed limited data. In addition, I have some major concerns as followings;
Major concerns
- Page8, Line281: The author should indicate body weights of HE or nTg mice or refer to the previous manuscript. Information of the body weights are important to think over the mouse behavior and muscle mass.
- Page10, Line336: For Figure 2, the author should add statistical data (E.g., signal intensity and area) about αBGT and synaptophysin staining.
- Page10, Line336: Does the HE mice has affected axons or/and de-nerved neuromuscular junctions? The author should show the axons of motor neurons in the muscle using antibody against e.g., peripherin.
- Page13, Line404: How about the muscle fiber size (Cross sectioned area) in the mouse mutant, CHMP2Bintron5? Is the factor changed?
- Page14, Line420: Skeletal muscle from ALS mouse models (e.g., TDP-43 mutations) and patients with ALS have disequilibrium in mitochondrial function (PMID: 11850111) and satellite cell activity (PMID: 25079897) as well as atrophy and neuromuscular junction degeneration. How about mitochondrial function and satellite cell activity in the mouse mutant, CHMP2Bintron5?
Minor concerns
- All legends of the Figures: Grammatically there should be a space before and after the characters, such as “=” and “<”; however, all figure legends often showed these characters incorrectly.
- Page2, Line70: There is a strange line break
- Page 7, Line269: Readers can follow easily, if the author indicates an approved gene symbol as well as their own expression; for example, Chrna1 as well as AchR α
- Page8, Line289: There is larger space between words, “8.5%” and “at”
- Page9, Line312 to 319: The author should refer to the results in Figure 1C and Figure 1D in the main text.
- Page 9, Line316-318: The author should add references, which shows increment of gene expressions for agrogin-1 and AChR in deserved muscle.
Author Response
Dear reviewer,
please find below the point-by-point answers we make to your comments.
MAJOR CONCERNS:
- Page8, Line281: The author should indicate body weights of HE or nTg mice or refer to the previous manuscript. Information of the body weights are important to think over the mouse behavior and muscle mass.
Reply: Theses data are now added and presented in figure 1C along with th grip strength and the muscle mass.
- Page10, Line336: For Figure 2, the author should add statistical data (E.g., signal intensity and area) about αBGT and synaptophysin staining.
Reply: As requested by the reviewer, based on the intensity of fluorescent signals, we have: 1- quantified the area covered by total and high intensity synaptophysin staining and aBGT staining; 2- calculated the ratios total synaptophysin area/aBGT area and high intensity synaptophysin area/aBGT area. These results are presented in figure 4 and discussed.
- Page10, Line336: Does the HE mice has affected axons or/and de-nerved neuromuscular junctions? The author should show the axons of motor neurons in the muscle using antibody against e.g., peripherin.
Reply: To answer this question, we have used tissues from YFP-HE mice we had. In these mice, YFP is highly expressed in motor neurons allowing the direct visualization of MN endings. As shown, axons clearly arrive in contact of the NMJ and fluorescence overly, either fully or partially, the post synaptic end plate. These data presented in figure 3 and discussed show the absence of retraction of nerve endings with complete disconnection and support a rather partial denervation.
- Page13, Line404: How about the muscle fiber size (Cross sectioned area) in the mouse mutant, CHMP2Bintron5? Is the factor changed?
Reply: to answer this point, CSA of SDH+ and SDH- fibers area have been measured. Results are presented in figure 7. Text and legends have been modified accordingly.
- Page14, Line420: Skeletal muscle from ALS mouse models (e.g., TDP-43 mutations) and patients with ALS have disequilibrium in mitochondrial function (PMID: 11850111) and satellite cell activity (PMID: 25079897) as well as atrophy and neuromuscular junction degeneration. How about mitochondrial function and satellite cell activity in the mouse mutant, CHMP2Bintron5?
Reply: Using RTqPCR, we have measured the mRNA levels of genes related to mitochondrial functions (one transcription factor controlling mitochondrial transcription and 3 genes involved in electron transport chain and 1 which is part of ATPsynthase). As shown in figure 8, expression of these genes increases with age in HE mice supporting the change seen in gene associated to the metabolism in the myofibers (also supported by the change in ATPase staining). These results are now included in the discussion.
We have also measured makers of satellite cells but the results obtained were not consistent and are not include in the manuscript.
Minor concerns
- All legends of the Figures: Grammatically there should be a space before and after the characters, such as “=” and “<”; however, all figure legends often showed these characters incorrectly.
- Page2, Line70: There is a strange line break
- Page 7, Line269: Readers can follow easily, if the author indicates an approved gene symbol as well as their own expression; for example, Chrna1 as well as AchR α
- Page8, Line289: There is larger space between words, “8.5%” and “at”
- Page9, Line312 to 319: The author should refer to the results in Figure 1C and Figure 1D in the main text.
- Page 9, Line316-318: The author should add references, which shows increment of gene expressions for agrogin-1 and AChR in deserved muscle.
Reply: All these points have been taken into account and corrected.
We hope that with these changes we have adequately answered to your the very constructive criticisms and feel that we have substantially improved our manuscript.
Best regards
Frédérique RENE
Reviewer 2 Report
The authors have previously generated mutant mice with neuron-restricted expression of the CHMP2Bintron5 that displayed hallmarks of ALS/FTD (Vernay et al., HMG 2016). CHMP2B gene is associated with both ALS and FTD, and the protein complex may play a role in vesicles biogenesis, trafficking, dendritic spines formation, and autophagy. The current manuscript used this mouse model to focus on the alteration of the neuromuscular junction (NMJ) and muscle metabolism with behavioral, histological, EMG and biochemical approaches. The authors have demonstrated that neuronal expression of CHMP2Bintron5 is sufficient to induce motor behavioral deficits, structural and functional changes in the NMJ, as well as a switch from fast-twitch glycolytic muscle fibers to more oxidative slow-twitch muscle fibers.
Strengths. Waegaert et al. did a thorough job in using multiple and comprehensive approaches to reinforce the importance of the NMJ in ALS pathogenesis and as a potential target for future therapeutic development. Another major strength of this manuscript is the demonstration of changes in muscle metabolism, which seems to be associated with the changes in NMJ morphology and function in CHMP2Bintron5 mutant mice. In addition, this manuscript further demonstrated that CHMP2Bintron5 mutant mice could be a useful model for ALS/FTD, as first shown in their 2016 HMG paper.
Major Concerns. The authors state that it is not currently known “whether CHMP2Bintron5 mutation alters the functionality of the NMJ prior to its destabilization, and thereby induces changes in muscle properties…” (Line 91, Page 3). The authors wanted to address this question. However, this reviewer could not find studies showing the alteration of the NMJ functionality occurred prior to NMJ destabilization. For example, the authors observed morphological changes (LM and EM) at 18-24 months of age in mutant mice (Figs. 2, 3). EMG studies showed CMAP decrements, as well as fasciculations and fibrillations, but also at 18-24 months of age. They did show gait subtle changes at 6 months preceding grip strength and muscle mass changes (line 447). However, they did not provide data showing whether EMG defects occurred at an earlier age, prior to NMJ destabilization. Thus, the statement in Discussion, “these results show that neuronal expression of CHMP2Bintron5 induces morphological changes in the presynaptic element and leads to abnormalities in neurotransmission that precede motor axon withdrawal” (Line 508, Page 16) is misleading. The authors should provide studies with different time lines to show functional changes indeed precede motor axon withdrawal at denervated NMJs. Ideally, the authors should also perform quantal analysis, which is more sensitive than EMG, to demonstrate the deficit of transmitter release in the mutant NMJs at the time before NMJ morphological denervation. Alternatively, the authors should at least rephrase those aims and relevant sentences.
Another concern is that the current manuscript appears redundant in some data, given that the authors’ 2016 paper has already shown defects in motor behaviors (Fig. 10), NMJ structures (Fig. 11 C-E), EMG (Fig. 11 A), as well as denervation markers (Fig. 11B) in CHMP2Bintron5 mice. It would be informative to discuss and compare these two studies and to highlight especially the novelty and significance of the current work to mitigate this concern. They should cite the specific paper(s) when they stated, “We have previously shown that CHMP2Bintron5 transgenic mice…” (Line 21), or in similar statements throughout the manuscript.
Other questions/concerns below also need to be addressed.
- Line 29, delete the s from muscles
- Lines 146-155. Please cite key references for the motor behavioral parameters, such as hind paws spacing, regularity index, duty cycle etc.
- Line 307, Fig. 1 legend, please insert (D) in “denervation markers (D) were evaluated.”
- Lines 316-319. Please cite key references for atrogin-1 and AChR expression.
- 2. Did the authors also examine terminal (perisynaptic) Schwann cells in the mutant NMJs? It would also be useful to quantify the denervation patterns as they did in the 2016 HMG paper (Fig. 11), or at least comment and compare these two studies.
- 3 EM. The nerve terminal in nTg appears smaller than that in HE muscles. The authors may like to consider to replace the picture with a larger nerve terminal in nTg mice to match better with the concept. Please explain in Fig. 3 legend what * in HE indicates? Did the authors observe any NMJs in mutant muscles that show junctional folds without nerve terminals, the so-called “vacated NMJ” in their EM studies? Furthermore, it is highly desirable to provide quantitative data, such as synaptic vesicle density, the number of endosomes, autophagosomes, as shown in other papers.
- Fig 4A, calibration bars may be wrong as fasciculation potentials should be larger than fibrillation potentials.
- Line 383, comparable should be compared?
- Line 486, should cite their 2016 HMG paper ref. 32, but not, or in addition to, ref. 31.
Author Response
Major Concerns. The authors state that it is not currently known “whether CHMP2Bintron5 mutation alters the functionality of the NMJ prior to its destabilization, and thereby induces changes in muscle properties…” (Line 91, Page 3). The authors wanted to address this question. However, this reviewer could not find studies showing the alteration of the NMJ functionality occurred prior to NMJ destabilization. For example, the authors observed morphological changes (LM and EM) at 18-24 months of age in mutant mice (Figs. 2, 3). EMG studies showed CMAP decrements, as well as fasciculations and fibrillations, but also at 18-24 months of age. They did show gait subtle changes at 6 months preceding grip strength and muscle mass changes (line 447). However, they did not provide data showing whether EMG defects occurred at an earlier age, prior to NMJ destabilization. Thus, the statement in Discussion, “these results show that neuronal expression of CHMP2Bintron5 induces morphological changes in the presynaptic element and leads to abnormalities in neurotransmission that precede motor axon withdrawal” (Line 508, Page 16) is misleading. The authors should provide studies with different time lines to show functional changes indeed precede motor axon withdrawal at denervated NMJs. Ideally, the authors should also perform quantal analysis, which is more sensitive than EMG, to demonstrate the deficit of transmitter release in the mutant NMJs at the time before NMJ morphological denervation. Alternatively, the authors should at least rephrase those aims and relevant sentences.
Reply:
According to reviewer comments, we have modified the sentence as follow: “Currently, it is not known whether CHMP2Bintron5 mutation alters the functionality of the NMJ, and thereby induces changes in muscle properties as has been observed with mutants of other ALS-related genes. ». However, knowing that the transgene expression is limited to neurons and that changes in gene expression (figure 8) and cross section area of muscle fibers (figure 9) are already detectable at 6 months of age, one can speculate that functional changes at the presynaptic terminal are already effective before the detection of change measured by electrophysiological technics used in this study, and prior to physical denervation (by definition, denervation means loss of contact between the nerve terminal and the muscle end plate). The use of technics proposed by the referee would indeed allow us to demonstrate the deficit of transmitter release but we do not have the expertise and equipment to perform this type of analysis in-house thus we were not able to perform this experiment. However, this point is discussed in more detail the discussion section.
Other questions/concerns below also need to be addressed.
- Line 29, delete the s from muscles
Reply: done.
- Lines 146-155. Please cite key references for the motor behavioral parameters, such as hind paws spacing, regularity index, duty cycle etc.
Reply: References 35 and 36 are added.
- Line 307, Fig. 1 legend, please insert (D) in “denervation markers (D) were evaluated.”
Reply: Legend is changed according to the modified figure 1.
- Lines 316-319. Please cite key references for atrogin-1 and AChR expression.
Reply: references added.
- Did the authors also examine terminal (perisynaptic) Schwann cells in the mutant NMJs? It would also be useful to quantify the denervation patterns as they did in the 2016 HMG paper (Fig. 11), or at least comment and compare these two studies.
Reply: We did not examine the terminal schwann cells (which is a study as such). The results obtained in this study are now compared along the discussion with the results obtained with CHMP2Bintron5 mice previously published by us and others.
- The nerve terminal in nTg appears smaller than that in HE muscles. The authors may like to consider to replace the picture with a larger nerve terminal in nTg mice to match better with the concept. Please explain in Fig. 3 legend what * in HE indicates? Did the authors observe any NMJs in mutant muscles that show junctional folds without nerve terminals, the so-called “vacated NMJ” in their EM studies? Furthermore, it is highly desirable to provide quantitative data, such as synaptic vesicle density, the number of endosomes, autophagosomes, as shown in other papers.
Reply: The reviewer is right, in EM, the nerve terminals of HE mice examined seemed systematically larger the those of nTg mice. As mentioned now in the text (lines 424-425, 689-691) this could be due to an accumulation of abnormal endosomes, autophagosomes and electron-dense structures linked with altered endo-lysosomal function.
- Legend of figure 5 (previously figure 3) is corrected.
- We did not observe so-called “vacated NMJ” which support the absence of full physical disconnection of the NMJ.
- As indicated to the editor when we received the referees comments, for technical reasons, we were not able to perform an additional EM analysis in the allotted time. Indeed, to be statistically valid, quantitative electron microscopy analyses would require analyzing a larger number of samples per genotype and age. We had subcontracted the initial EM study to an external platform. After contacting them the plateform informed us that they won’t be able to process our request for new samples preparation before several months. These quantifications are thus not included in the revised manuscript.
- Fig 4A, calibration bars may be wrong as fasciculation potentials should be larger than fibrillation potentials.
Reply: calibrations were correct but the legend was wrong; Correction is made.
- Line 383, comparableshould be compared?
Reply: change made.
- Line 486, should cite their 2016 HMG paper ref. 32, but not, or in addition to, ref. 31.
Reply: change made.
We hope that with these changes we have adequately answered to your the very constructive criticisms and feel that we have substantially improved our manuscript.
Best regards
Frédérique RENE
Reviewer 3 Report
Waegaert et al. discussed that neuronal expression of CHMP2Bintron5 is sufficient to induce abnormal presynaptic morphology and NMJ transmission, which ultimately results in ALS-like phenotype, and consequently alter the properties of muscle fibers. This manuscript harbors some interesting information in this field, but requires more detailed analysis on the phenotypes to demonstrate the pathogenesis induced by the CHMP2Bintron5 mutation.
Specific comments
- Describe the CHMP2Bintron5 mutation more in detail.
- Through this manuscript, authors discussed the differences between non transgenic mice and the CHMP2Bintron5 transgenic mice. However, the effects of the WT CHMP2B should be considered. Therefore, authors should discuss the differences between WT CHMP2B transgenic mice and SHMP2Bintron5 transgenic mice.
- Denervation in HE mice should be demonstrated not only by the expression of denervation marker (AChR alpha) but by the analysis of immunohistochemistry showing the decrease in the number of NMJs. Moreover, in Figure 1D, authors should examine other denervation markers other than AChR alpha, such as AChR gamma.
- In Figure 2, immunohistochemical analysis of synaptophysin and muscle end-plate in HE mice at 18-24 months should be quantitatively presented. Preferably, time course analysis of the immunohistochemical analysis for synaptophysin and muscle end-plate should be presented to reveal the timing of the onset of the NMJ degradation.
- In Figure 3, quantitative analysis of synaptic vesicles, enlarged endosomes, post-synaptic folds, and autophagosomes in electron microscopy should be presented. Since AChR alpha is significantly increased at 12 months of age in HE mice, abnormal synaptic vesicles may have been observed even at the earlier stage of the mice. Thus, the analysis at the earlier stage should be presented.
- In Figure 5, fiber types switch should be demonstrated not only by the expression of fiber type specific markers, but also by the histological analysis showing the actual proportion of each fiber types.
- What is the causal relationship between fiber-type switching/metabolic change and the synaptic abnormality?
- In discussion, discuss more in detail about the molecular mechanisms underlying synaptic abnormality caused by the CHMP2Bintron5 mutation and its relationship to ALS pathology.
Author Response
Dear reviewer,
please find below the point-by-point answers we make to your comments.
Specific comments
- Describe the CHMP2Bintron5mutation more in detail.
Reply: A more detailed description is now included in the introduction (lines 84-93).
- Through this manuscript, authors discussed the differences between non transgenic mice and the CHMP2Bintron5 transgenic mice. However, the effects of the WT CHMP2B should be considered. Therefore, authors should discuss the differences between WT CHMP2B transgenic mice and SHMP2Bintron5 transgenic mice.
Reply: As far as we know, WT Chmp2B transgenic mice have only been used in two studies (see [31] and Clayton et al., 2014, 26358247). These mice expressing the WT form of CHMP2B do not develop any motor symptoms. This is now indicated in the text lines 610-611.
- Denervation in HE mice should be demonstrated not only by the expression of denervation marker (AChR alpha) but by the analysis of immunohistochemistry showing the decrease in the number of NMJs. Moreover, in Figure 1D, authors should examine other denervation markers other than AChR alpha, such as AChR gamma.
Reply: As previously published by us and others ([31, 32]) denervation evaluated by using synaptophysin IHC is very limited. In this study, we have examined the physical connection between the nerve terminal and the corresponding endplate using tissues from YFP-HE mice. In these mice, YFP is highly expressed in motor neurons allowing the direct visualization of MN endings. As shown, axons clearly arrive in contact of the NMJs and fluorescence covers, either fully or partially, the post synaptic endplate. These data presented in figure 3 and discussed show the absence of retraction of nerve endings with complete disconnection and support a rather partial denervation. In addition, as required, we have examined other denervation markers (AChRe, AChRg and musk). The results are now presented in figure 2 and discussed along with the changes in immunostainings and CMAPs (second paragraph of the discussion).
- In Figure 2, immunohistochemical analysis of synaptophysin and muscle endplate in HE mice at 18-24 months should be quantitatively presented. Preferably, time course analysis of the immunohistochemical analysis for synaptophysin and muscle endplate should be presented to reveal the timing of the onset of the NMJ degradation.
Reply: As requested by the reviewers 1 and 3, based on the intensity of fluorescent signals, we have: 1- quantified the area covered by total and high intensity synaptophysin staining and by aBGT staining; 2- calculated the ratios total synaptophysin area/ aBGT area and high intensity synaptophysin area/aBGT area. These results are presented in figure 4 and discussed.
- In Figure 3, quantitative analysis of synaptic vesicles, enlarged endosomes, post-synaptic folds, and autophagosomes in electron microscopy should be presented. Since AChR alpha is significantly increased at 12 months of age in HE mice, abnormal synaptic vesicles may have been observed even at the earlier stage of the mice. Thus, the analysis at the earlier stage should be presented.
Reply: As indicated to the editor when we received the referees’ comments, for technical reasons, we were not able to perform an additional EM analysis in the allotted time. Indeed, to be statistically valid, quantitative electron microscopy analyses would require analyzing a larger number of samples per genotype and age. We had subcontracted the initial EM study to an external platform. After contacting them the plateform informed us that they won’t be able to process our request for new samples preparation before several months. These quantifications are thus not included in the revised manuscript.
- In Figure 5, fiber type switch should be demonstrated not only by the expression of fiber type specific markers, but also by the histological analysis showing the actual proportion of each fiber types.
Reply: To address this point, we have performed myofibrillar ATPase staining on TA section at the three stages studied. Results are now presented in figure 9, 10 and supplementary figure S6, and discussed.
- What is the causal relationship between fiber-type switching/metabolic change and the synaptic abnormality?
- In discussion, discuss more in detail about the molecular mechanisms underlying synaptic abnormality caused by the CHMP2Bintron5 mutation and its relationship to ALS pathology.
Reply: these two points are now discussed in more detail in the last part of the discussion (742-767).
We hope that with these changes we have adequately answered to your the very constructive criticisms and feel that we have substantially improved our manuscript.
Best regards
Frédérique RENE
Round 2
Reviewer 1 Report
The authors have improved manuscript quality and data quality. I also have no serious criticisms regarding methodology and interpretation of results.
Reviewer 3 Report
Most of the concerns are now appropriately addressed.